# Role of Cadherins in Cancer—A Review

**DOI:** 10.3390/ijms21207624

**Published:** 2020-10-15

**Authors:** Ilona Kaszak, Olga Witkowska-Piłaszewicz, Zuzanna Niewiadomska, Bożena Dworecka-Kaszak, Felix Ngosa Toka, Piotr Jurka

**Affiliations:** 1Department of Small Animal Diseases, Institute of Veterinary Medicine, Warsaw University of Life Sciences, 02-787 Warsaw, Poland; piotr_jurka@sggw.edu.pl; 2Department of Pathology and Veterinary Diagnostics, Institute of Veterinary Medicine, Warsaw University of Life Sciences, 02-787 Warsaw, Poland; 3Carnivore Reproduction Study Center, Ecole Nationale Veterinaire d’Alfort, 94700 Maison Alfort, France; zuzanna.niewiadomska@vet-alfort.fr; 4Department of Preclinical Sciences, Institute of Veterinary Medicine; Warsaw University of Life Sciences, 02-787 Warsaw, Poland; bozena_kaszak@sggw.edu.pl; 5Center for Integrative Mammalian Research, Ross University School of Veterinary Medicine, BOX 334 Basseterre, Saint Kitts and Nevis, West Indies; ftoka@rossvet.edu.kn

**Keywords:** cadherin, cancer signaling, cell-cell adhesion, cancer progression

## Abstract

Cadherins play an important role in tissue homeostasis, as they are responsible for cell-cell adhesion during embryogenesis, tissue morphogenesis, differentiation and carcinogenesis. Cadherins are inseparably connected with catenins, forming cadherin-catenin complexes, which are crucial for cell-to-cell adherence. Any dysfunction or destabilization of cadherin-catenin complex may result in tumor progression. Epithelial mesenchymal transition (EMT) is a mechanism in which epithelial cadherin (E-cadherin) expression is lost during tumor progression. However, during tumorigenesis, many processes take place, and downregulation of E-cadherin, nuclear β-catenin and p120 catenin (p120) signaling are among the most critical. Additional signaling pathways, such as Receptor tyrosine kinase (RTK), Rho GTPases, phosphoinositide 3-kinase (PI3K) and Hippo affect cadherin cell-cell adhesion and also contribute to tumor progression and metastasis. Many signaling pathways may be activated during tumorigenesis; thus, cadherin-targeting drugs seem to limit the progression of malignant tumor. This review discusses the role of cadherins in selected signaling mechanisms involved in tumor growth. The clinical importance of cadherin will be discussed in cases of human and animal cancers.

## 1. Introduction

Cadherins are transmembrane glycoproteins responsible for cell-cell adhesion and maintenance of normal tissue architecture. In addition, cadherins have been found not only between the tumor cells but also in body fluids (mainly in blood) [1,2,3]. The classical cadherins, among which there are many subgroups, are a class of adhesion molecules that interact with catenins through the cytoplasmic domain [1]. The classical cadherins are epithelial-cadherin, placental-cadherin and neural-cadherin (E-, P- and N-cadherin, respectively).

The structure of cadherin is well known and has been extensively reviewed by others [2,3,4,5]. Briefly, the extracellular part of classical cadherin is built of five extracellular cadherin domains (ECs). There are at least 20 subtypes of the classical cadherin encoded in the mammalian genome, and all of them are similarly organized as the 5-EC structure. The five cadherins mediate Ca-dependent adhesion via their extracellular domains provide homotypic cell-cell interactions, and a cytoplasmic tail binds to several adaptor molecules to transmit physical and biochemical signals to the cell [6]. The frequent cytoplasmic binding partner is β-catenin but certain cadherins may also bind the plakoglobin [7]. Cadherins extracellular domains join with cytoplasmic tail to form signaling hubs called adherens junctions (AJs). Through AJs, cadherin interacts with cadherin on the adjacent cells, forming a zipper-like structure. The AJs connect the actin cytoskeleton of neighboring cells through direct interaction. Besides maintenance of tissue architecture, crucial stages of embryogenesis such as formation of the gastrula, neurula and organogenesis rely on the expression of cadherin. Moreover, progesterone with endometrial calcitonin regulates the expression of cadherin, which has an impact on endometrial tissue compactness and embryo implantation [8]. Later, in mature tissues, cadherins are responsible for the maintenance of cell polarity, tissue integrity and homeostasis [9]. However, during carcinogenesis, cadherins are often inactivated or functionally blocked, allowing the development and progression of cancer or the metastatic processes [6]. 

The role of cadherins in the process of cancer development has been studied widely over the last decades [6,10,11,12]. In human medicine, cadherins have been evaluated in many malignancies, such as pancreatic cancer, melanoma, hepatocellular carcinoma, glioblastoma, breast and gastric cancers [13,14,15,16]. Because most of humans’ solid tumors are of epithelial origin, adhesion molecules at the junctions of epithelial cells and cell signaling pathways are of great interest. Many studies have described E-cadherin function as a tumor suppressor [17,18,19]. However, recent studies have shown that E-cadherin, especially in late-stage cancers, may also promote cell migration, invasion and even tumor progression [20,21,22,23,24]. Genetic deficits in E-cadherin cell surface regulation may contribute to cancer development [25]. Some studies have described a process which occurs during metastatic colonization named mesenchymal-to-epithelial transition (MET), in which E-cadherin is re-expressed [26]. E-cadherin has also been shown to promote metastasis in diverse models of invasive ductal carcinomas [27]. Overexpression of N and P-cadherin in breast cancer patients is also a common finding, and it is usually related to poor prognosis [10]. The detailed role of N and P-cadherins will be discussed in further paragraphs.

The purpose of this review is to discuss the role of cadherins in the tumor growth and its clinical importance.

## 2. Cadherin-Catenin Complex

Cadherins preferentially bind to catenin proteins that are found in three subtypes, α-catenins, β-catenins, and p120ctn family catenins. A juxtamembrane region of cadherins, which is proximal to the cell membrane, binds to p120ctns, while the distal region called catenin-binding domain binds to β-catenins. Monomeric forms of α-catenins associate with the cadherin-catenin complex through β-catenins, while homodimeric forms of α-catenins do not bind β-catenins, but F-actin [28]. The connection between the cadherin and catenins has been referred as the cadherin-catenin complex. Cadherins, through their highly conserved cytoplasmic domains, bind to catenins; thus, they mediate biochemical signals (Figure 1). Specifically, the juxtamembrane domain of cadherin cytoplasmic tail binds to a family of proteins, including p120 catenin (p120; CTNND1), neural plakophilin-related armadillo protein (NPRAP/δ-catenin; CTNND2), armadillo repeat protein deleted in velo-cardio-facial syndrome (ARVCF) and plakophilin 4 (p0071). This connection is essential for the stabilization and function of the classical cadherin binding though other proteins to microtubules and kinesins, which regulates cadherin localization. As mentioned before, cadherins also often bind to β-catenin (CTNNB1) through the catenin-binding domain, which is located at the distal region of cadherin cytoplasmic tail. The interaction between β-catenin and the cytoplasmic domain of cadherin enables clustering into junctional structures. Any dysfunction of the cadherin-catenin complex reduces cell adhesion. Such dysfunction of the cadherin-β-catenin complex was seen in the neoplastic process leading to β-catenin accumulation in the cytoplasm and nucleus of the tumor cells. The function of β -catenin is equally important, because its association with cadherin links the Ajs to filamentous actin and promotes the re-organization of the actin cytoskeleton. Catenins mediate many signaling processes, which functionally connect cadherin-catenin complex to the cytoskeleton. Thus, they control the balance between cell-cell adhesion and cell differentiation, and regulate cell growth and motility [17]. The cadherin-catenin complexes are crucial for the ability of the cells to adhere to each other.

## 3. Cadherin and Catenin Signaling in Cancer

Cadherins, together with catenins, interact through the AJs with actin cytoskeleton and regulate cellular processes during embryonic development as well as during neoplastic transformation and progression. In cancer, any disturbances in cell-cell and cell-matrix adhesion are related to tumor progression and allow cancer cells to become more motile, degrade the extracellular matrix, enter the blood vessels and form distant metastases [9,30]. 

The signaling activation takes place at the site of cadherin-catenin interaction. Thus, the catenin-binding domain of cadherin is crucial in cadherin function and it plays an important role in maintaining epithelial integrity. For example, phosphorylation of either E-cadherin or β-catenin affects β-catenin binding to cadherin, while phosphorylation of β-catenin may also lead to α-catenin binding to the cadherin-β-catenin complex [17]. During tumorigenesis, due to the phosphorylation, β-catenin is released into the cytoplasm (also as a consequence of E-cadherin loss) and communicates with a protein complex composed of Axin, adenomatous polyposis coli (APC) and the Ser/Thr kinase glycogen synthase 3β (GSK-3β) [17,31]. The phosphorylation of β-catenin is induced by GSK-3 β as well as casein kinases I and II and leads to ubiquitination and subsequent degradation in the 26S proteasome. The E-Cadherin/β-catenin complex also affects the Wnt-signaling pathway (the vertebrate homolog of Wingless). Phosphorylated β-catenin enters in the degradation pathways unless WNT signaling is activated. β-catenin is considered the prime signal transducer of canonical Wnt pathway. Wnt pathway activation contributes to translocation of intact β-catenin to the nucleus, where, together with the lymphoid enhancer factor (LEF)/T-cell factor (TCF), it activates a variety of transcription factors, increasing transcription of target genes, such as *Fz, LRP6, Axin2, Naked, Dkk1*, and *Rspo*, resulting in positive or negative regulation by TCF/β-catenin [17,29,32,33,34,35,36,37]. Therefore, tyrosine phosphorylation of β-catenin leads to beta-catenin signaling activation (and transcriptional impact), whereas β-catenin degradation inhibition in the presence of Wnt signaling is an inactivating mechanism.

Furthermore, p120-catenin, which interacts with the cadherins through the juxtamembrane domain is regulated by tyrosine kinases and modulate cadherin intracellular trafficking, stability, adhesive capacity and motility [29]. p120 can be phosphorylated at multiple serine, threonine and tyrosine residues. The phosphorylation at many tyrosine sites (Y) is mediated by Src family kinases within its N-terminus, including Y96, Y112, Y228, Y257, Y280, Y291, Y296 and Y302. The p120 phosphorylation at serine/threonine sites regulates E-cadherin expression at the cell membrane, while p120 phosphorylation of tyrosine and serine sites influences on the strength of cadherin activation and adhesion [38]. p120-catenin may also influence cell adhesion by modulating the organization of actin cytoskeleton by activation of RhoA [29]. Many processes, including intracellular signaling pathways such as Wnt, transforming growth factor β (TGF-β), mitogen activated kinase (MAPK), gene transcription, protein stability and posttranslational modifications, can affect cell adhesion [29]. For example, TGF-β is a growth inhibitor of many cell types and E-cadherin enhances TGF-β signaling, and therefore, suppresses uncontrolled cell proliferation [39]. E-cadherin also activates MAPK signaling through epidermal growth factor receptors activation [40]. MAPK is an important signaling pathway, as it controls cell proliferation, differentiation and survival. Changes in the cell adhesion are important for tissue morphogenesis and repair processes but may also be related to tumor development. There are many oncogenic signaling pathways related to cadherin cell-cell adhesion. Among them are cyclin kinase inhibitor p27-mediated, phosphatidylinositol-3 kinase (PI3K)/AKT and ras-related C3 botulinum toxin substrate (Rac1) signaling, rat sarcoma viral oncogene (Ras), MAPK, as well as Hippo signaling [41,42,43,44]. For example, E-cadherin can increase the stability of cyclin-dependent kinase inhibitor p27, thereby upregulating p27 expression and inhibiting proliferation. Therefore, E-cadherin is also considered a growth suppressor [41,43]. Signaling by class 1 PI3-kinases is activated in response to various extracellular stimuli [45]. Such cadherin-activated PI3-kinase signaling has an impact on cadherin function and strength of cell-cell adhesion. E-cadherin mediated cell adhesion formation of tight junctions and apical polarity complexes leads to the Hippo signaling pathway activation. This pathway may be inhibited by mechanical stress; however, increased inhibition upon cell contact may be triggered by its activation [46]. However, the most common pathway of tumor development and progression is the nuclear β-catenin/T-cell factor (TCF) signaling pathway [17]. The cadherin-catenin complex can act both as a potentiator and attenuator of Wnt/β-catenin signaling [47]. In the first instance, cadherin-based cell-cell adhesion could increase the amino-terminal phosphorylation of β-catenin and its subsequent rate of destruction. In the second instance, cadherin-catenin complex is required for Wnt/β-catenin signaling, as it recruits kinases needed for the execution of canonical Wnt signals [47]. All these various mechanisms and the loss of anti-tumorigenic E-cadherin signaling, together with the presence of nuclear catenin signaling and the activation of various additional pathways (RTK, RhoGTPase), are main events related to tumor progression. 

Many studies have shown the role of β-catenin as a proto-oncogene in human cancer [17,31]. Mutations in the gene CTNNB1 that encodes β-catenin have resulted in cancer. It is estimated that at least 10% of cancer samples sequenced have exhibited mutation in CTNNB1 [48]. These mutations were mostly located in the N-terminal region of β-catenin, β:TrCP binding motif. Damage to this binding motif disables ubiquitination and degradation of β-catenin. As such, stabilized β-catenin accumulates in the cytoplasm and then is translocated to the nucleus, where it concomitantly drives transcription of its target genes [49]. Apart from its role in tumor progression, the nuclear signaling pathway of β-catenin is responsible for the pluripotent phenotype and self-renewal of both normal and cancer stem cells [17,41,49]. 

## 4. Epithelial Mesenchymal Transition

Epithelial-mesenchymal transition (EMT) is a pivotal process of morphogenesis whereby epithelial cells acquire the mesenchymal phenotype. EMT is known as the phenomenon of loss of E-cadherin expression during cancer progression [50]. EMT occurs during embryonic development processes such as gastrulation, neural crest development and placenta formation. Because EMT is an essential hallmark in embryogenesis, scientists postulated the idea that epithelial cancers must acquire mesenchymal characteristics for invasion and metastasis [51,52]. 

In carcinomas, EMT can be initiated and promoted by many oncogenic signaling pathways, including hypoxia and signals of the tumor microenvironment (Figure 2). Subsequently, this leads to epithelial cells losing their cell polarity and cell–cell adhesion, thus gaining the migratory and invasive properties [50]. In breast cancer, a “cadherin-switch” is defined as above-mentioned loss of E-cadherin and increased expression of N-cadherin during tumor progression [53]. This transition induces or enhances the metastatic capacity of the carcinoma cells. There are many identified mechanisms of E-cadherin mediated tumor suppression [54,55,56]. The mechanism of tumor suppression may come from the adhesive function of E-cadherin, which prevents tumor cells from dissociating from one another, and therefore, prevents their migration to other tissues. The alternative mechanism of tumor suppression is via antagonizing the nuclear signaling function of β-catenin and altering its ability to regulate target genes that support tumor invasion, as β-catenin is a proto-oncogene [54]. It was shown that only E-cadherin constructs that had β-catenin binding were able to retain the tumor growth and not E-cadherin constructs that exhibited adhesive activity without β-catenin binding. In addition, E-cadherin-catenin complex possesses the ability to downmodulate NF-*κ*B activity, which leads to inflammation-associated carcinogenesis [57]. There is also the epigenetic mechanism controlling the E-cadherin action directly associated with expression of microRNAs, microRNA-9 and microRNA-10b [58]. A recent study described tumor derived exosomes that tend to transfer EMT-related RNAs and proteins to recipient cells, mediate the instability of cadherins, and promote cancer progression [59]. The other tumor suppressing mechanism by E-cadherin complexes may be associated with the RNA interference (RNAi) machinery at epithelial AJs [60]. However, these novel findings require further examination. 

Cadherin switching is primarily the result of transcriptional regulation of cadherin expression through several factors, such as SNAI1, SLUG, TWIST, ZEB1 and ZEB2 [61,62]. These factors act through repression of E-cadherin transcription by directly binding to its promoter [63]. The main transcriptional activation of EMT are the Wnt/β catenin, TGF-β and Hedgehog signaling pathways [62,63,64,65]. In addition, signaling molecules such as ILK, FAK and SRC have an impact on the activation of EMT [66]. TGF- β, in co-operation with Ras, induces Snail and Slug, which then leads to the disruption of desmosomes followed by cell motility, inhibition of cytokeratin expression and increase in vimentin expression. Finally, cell spreading occurs leading to separation of cell-cell attachment [67,68,69]. The Wnt/catenin and TGF- β signaling pathways in EMT process are quite similar and have been shown in breast cancer bone metastasis [70].

The protein p120 catenin is another important factor that plays a role in the regulation of downstream signaling and cell behavior upon cadherin switch [55,71]. p120 is encoded by CTNND1 gene, and functions to maintain the stability and turnover of E-cadherin as well as regulate Rho GTPases [72]. The loss or inactivity of E-cadherin leads to translocation of p120 to the cytosol, where p120 now can control the invasiveness of tumor cells [72,73]. It activates the Rac1-MAPK (mitogen-activated protein kinase) signaling pathway and promotes transformed cell growth [55]. Moreover, when E-cadherin expression switches to P-cadherin, further cytosolic translocation of p120 occurs leading to induction of tumor cell migration through activation of RAC1 and CDC42 [74,75], tumor anchorage-independence and increased tumor growth in vivo [76]. The tumor invasiveness and cell migration are increased by the upregulation of mesenchymal cadherins and the downregulation of E-cadherins in tumors of epithelial origin. However, pro-tumorigenic cell behavior may be triggered by the expression of mesenchymal cadherins even in the presence of E-cadherin [77].

In humans, the loss of E-cadherin expression during EMT is associated with tumor development. However, decreased expression of E-cadherin has also been observed in canines with more aggressive mast cell tumors (Grade 3) [78]. Similarly, the change in expression of E-cadherin to N-cadherin or the up-regulation of cadherin-11 and P-cadherin also favor tumor progression and are an indicator of more migratory and more invasive tumor types [11]. In many studies, it was shown that high expression of E-cadherin was associated with prevention of tumor progression into a more invasive type [55,79]. Therefore, E-cadherin up-regulation can be associated with inhibition of tumor cell invasion, and can also be considered a tumor suppressor [17,19]. 

## 5. Cadherins in Human and Animal Cancer as a Prognostic Factor

Cadherin expression has been widely studied in human cancers as well as in animal models [4,10,21]. Expression of different cadherins in cancer is diverse and multifactorial. 

P-cadherin is expressed in myoepithelial cells, and its overexpression in high-grade tumors is correlated with tumor aggressiveness and a poor prognosis in humans [4]. Similarly, in canine, malignant mammary tumors anomalous expression of P-cadherin was associated with invasion [81]. It was demonstrated that in cats, P-cadherin expression is significantly related to high histological grade of carcinomas, tumor necrosis, infiltrative growth and presence of neoplastic emboli [82]. However, in human squamous cell carcinoma of the oral cavity (OSCC), the reduced expression of P-cadherin and E-cadherin contributed to the invasive potential of early OSCC and was associated with poor prognosis [83]. According to Kim et al., in gastric carcinomas the expression of P-cadherin was intact, while no expression of P-cadherin was detected in non-neoplastic gastric mucosal tissues from gastric carcinoma patients. Additionally, the P-cadherin gene from gastric carcinoma was hypomethylated in its promoter region, whereas the same gene in non-carcinoma cells was methylated. Such presentation of P-cadherin expression was associated with favorable prognosis [84]. On the other hand, in vitro experiments on a pancreatic ductal adenocarcinoma (PDAC) cell line that does not express P-cadherin showed that ectopic expression of P-cadherin (Panc1-CDH3) allowed the PDAC cells to acquire increased motility, which, however, could be blocked by an antibody against P-cadherin. These data provide the evidence that overexpression of P-cadherin is likely related to the biological aggressiveness of PDACs [85]. A closer examination of the prognostic value of the cadherin switch during bladder cancer shows that while the decrease of E-cadherin and the gain of N-cadherin gene expression represented risk factors for cancer-related death, the expression of P-cadherin proved to be a significant independent prognostic factor for both cancer-specific and recurrence-free survival [86]. In pursuit of potential diagnostic usefulness of P-cadherin, Riener et al. [87] used immunohistochemistry on tissue microarrays to evaluate carcinomas and dysplastic lesions of the biliary tract. They detected P-cadherin in most of extrahepatic cholangiocarcinomas (73%) and gallbladder carcinomas (64%) and in some of intrahepatic cholangiocarcinomas (37%). Dysplastic biliary epithelium highly expressed P-cadherin. In those studies, P-cadherin was detected at early stage of carcinogenesis and was, therefore, thought to be a useful marker for the early detection of cholangiocarcinoma. In yet another study involving 69 patients with hepatocellular carcinoma, over half of the liver samples exhibited downregulation of P-cadherin compared to primary human hepatocytes and non-malignant liver tissue. In these cases, reduced expression of P-cadherin induced tumorigenicity and was deemed as a prognostic marker of this highly aggressive hepatocellular carcinoma [88]. In summary, expression P-cadherin appears to be dependent on the type of cancer, and therefore, its prognostic value cannot be generalized.

N-cadherin (the product of the CDH2 gene) influences the nervous system, brain, heart, skeletal muscles, blood vessels and hematopoietic microenvironment function [89,90,91]. N-cadherin is overexpressed in invasive and metastatic breast cancer and induces metastasis by potentiating signaling by the FGF receptor in humans [10]. There are various factors which modulate N-cadherin expression in tumor cells such as TGF-β1, Wnt/β-catenin, EGFR and NF-κB. Abnormal expression of N-cadherin has also been found in many other cancers, such as lung cancer, hepatic cancer, urothelial cancer and prostate cancer. [89,90,91,92,93,94]. In tumor progression, it has been documented that abnormal expression of N-cadherin is connected with malignancies manifested by cell transformation, apoptosis, angiogenesis, invasion and metastasis [89]. Conversely, in canine choroid plexus tumors, N-cadherin immunolabeling was more expressed in grade I tumors [95]. In feline adenomas and carcinomas, N-cadherin expression is associated with a reduced expression of E-cadherin and the presence of regional metastasis [96]. High presence of N-cadherin in colorectal cancer significantly associated with tumor differentiation, tumor size, tumor nodes and metastasis stage. Lower overall survival and disease-free survival rate was characteristic for patients with high N-cadherin expression compared to patients who had low N-cadherin expression. According to this study, high N-cadherin expression in colorectal cancer was an independent prognostic factor [97]. Lascombe et al. [98] reported that normal urothelium did not express N-cadherin but observed increased expression in advanced stage superficial urothelial tumors and proposed N-cadherin as a novel prognostic marker of progression in superficial urothelial tumors. It appears that expression of N-cadherin may have varied significance depending on the tumor type. For instance, a physiological pattern of N-cadherin expression was observed in renal cell carcinoma (RCC) specimens from patients undergoing surgery, despite a high tumor grade, and patients with RCC and normal N-cadherin-expression had a poorer prognosis than those with N-cadherin-abnormal RCC [99]. Therefore, this type of cadherin may play a different role from that of E-cadherin and may be associated with the aggressiveness and malignant potential of RCC.

The most frequently evaluated cadherin is E-cadherin [20]. This adhesion molecule is an important component of the apical zonula adherents in the epithelial monolayers and is a regulator of the epithelial structure [17,100,101]. A recent meta-analysis of breast cancer patients carried out by Li et al. [102] revealed that reduction of E-cadherin on tumor cells was significantly associated with poorer overall survival and disease-free survival, and correlated with clinicopathological features such as tumor size, lymph node status, TNM stage and histological grade [18,103]. E-cadherin expression has also been studied in animal models. However, in veterinary species, the involvement of E-cadherin in cancers is just beginning to be unraveled. For instance, in mammary gland tumors of bitches, decreased expression of E-cadherin is associated with tumor malignancy and metastatic progression, which leads to shorter patient survival [103,104]. Zuccari et al. showed low expression of E-cadherin in canine mammary neoplasia and considered it as a marker of favorable prognosis. However, when cachexia and obesity in affected dogs occurred with low E-cadherin expression, an unfavorable prognosis was obvious [105]. The conflicting results suggest that more studies on this field should be done. On the other hand, a study of 93 dogs with canine cutaneous histiocytoma (CCH) revealed that the lack of E-cadherin expression on tumor cells might indicate an activation or maturation process of the tumor cells accounting for a switch to CCH regression [106]. A similar expression pattern of E-cadherin has been detected during carcinogenesis of the canine prostatic epithelial cells, [107], canine and feline meningiomas [108] and ovine intestinal adenocarcinoma [109]. Because of some similarities in the behavior of cadherin, animal tumors should be explored as comparative models for studies in human cancers to better understand the role of E-cadherin in tumorigenesis. Therefore, low-expression of E-cadherin can stand in as a predictor of poorer prognosis and could be a valuable therapeutic target for breast cancer patients.

In dogs, reduction in E-cadherin expression, like humans, is associated with increased tumor size, high histological and invasion grades, lymph node metastasis and a high mitotic index [21]. Another study has demonstrated that the expression of classical cadherins is altered during tumor progression in feline neoplasms [96]. In addition, E-cadherin in feline mammary tumors demonstrated its reduction or absent expression in carcinomas when compared to benign lesions [110]. On the other hand, the prognostic value of E-cadherin in feline carcinomas is still unknown. Regarding the prognostic value of E-cadherin in different types of cancer, in a large series of RCC with tumor thrombus (TT) of vena cava (VC), increased expression of E-cadherin was connected with initial lymph node metastasis and with both worse OS (overall survival) and worse CSS (cause-specific survival). Such presentation of E-cadherin may aid in identifying recurrence risk patients in whom adjuvant therapy could be beneficial. E-cadherin overexpression in sarcomas reduces anchorage-independent growth and spheroid formation of sarcoma cells through downregulation of phosphorylated CREB1 (p-CREB) and the transcription factor, TBX2, thus inhibiting sarcoma aggressiveness by preventing anchorage-independent growth [111]. In metastatic colorectal cancer (mCRC), expression of E-cadherin in either cell membrane or cytoplasm was combined with strong vascular endothelial growth factor A (VEGF-A) staining as a predictor of disease outcome. VEGF-A expression was significantly connected with E-cadherin expression in the cytoplasm. Both enhanced E-cadherin expression in the cytoplasm and decreased expression of E-cadherin in the cell membrane indicate a poor prognosis in mCRC [112]. Besides the protein expression levels of E-cadherin, the genetic variation of CDH1 gene in dogs appears to be important. Of the three known single nucleotide polymorphisms (SNP) of CDH1, rs850805755, rs852280880 and rs852639930, rs850805755 and rs852280880 were associated with a decreased risk and a later onset of mammary tumor development. Furthermore, these SNPs were characterized by small size carcinomas, low histological grade and low nuclear pleomorphism. On the hand, SNP rs852639930 was characterized by a non-infiltrative, non-invasive growth pattern and development of small size tumors. Therefore, these SNP variants may indicate a low tumor development [113]. 

## 6. Therapeutic Targets Associated with Cadherin Dysfunction 

Understanding how cadherins influence the cell behavior can be used to design possible therapeutic interventions to regulate its activity and prevent tumor cell growth, invasion and metastasis [12,114]. 

A good example is α-solanine (a glycoalkaloid extract of *Solanum nigrum* Linn.), which stimulates E-cadherin expression and reduce vimetin expression (mesenchymal marker) leading to the suppression of EMT. At the same time α-solanine downregulates matrix metalloproteinase (MMP) expression, which plays a key role in metastatic process; therefore, it is has a strong anti-carcinogenic activity [115]. Additionally, drugs such as simvastatin and metformin upregulate E-cadherin and inhibit N-cadherin in human prostate cancer cells, contributing to blockade of TGF-β1-induced EMT [116,117].

Many synthetic peptides are used in cancer therapy. In recent studies ADH-1 synthetic cyclic peptide, which mimics the natural HAVD sequence of N-cadherin, has been used in myeloma, neuroblastoma and pancreatic cancer in vitro [118,119,120]. It leads to inhibition of angiogenesis, metastasis, cell proliferation and tumor growth. A murine pancreatic cancer model tested the therapeutic potential of ADH-1, an antagonist of N-cadherin. Results showed that ADH-1 has significant antitumor activity against N-cadherin–expressing pancreatic cancer cells [120]. Human clinical trials also confirmed the antitumor activity of ADH-1 in gynecological cancers [121]. Other N-cadherin antagonists, synthetic linear peptide H-SWTLYTPSGQSK-NH 2 blocked neurite outgrowth, myoblast fusion and cell migration in breast cancer in vivo [122]. In addition, monoclonal antibodies can be used in cancer therapy. Antibodies against N-cadherin inhibit the metastases and suppress tumor growth in prostate cancer in vivo [123].

Other therapeutic targets may be epigenetic activation of E-cadherin or inactivation of N-cadherin. In different types of cancer increased invasiveness is connected with downregulation of the E-cadherin-encoding gene (CDH1). In human breast, hepatocellular and prostate cancer hypermethylation of the CDH1 promoter has been observed [124,125]. It was documented that decreased DNA methylation leads to inhibition of tumor growth in a mouse model for colorectal tumor; thus, this can be a novel target for anticancer therapy [126]. For instance, thymine-DNA-glycosylase (TDG) has been directly targeted to specific sequences in the DNA to produce local DNA demethylation at critical regulatory sequences and lead to enhanced gene induction [125]. Another idea is to target an intramembrane protease of the Rhomboid family-RHBDL2 to control cancer cell migration by E-cadherin functional inactivation [127]. As mentioned earlier, activation of Wnt/β-catenin signaling is important for the initiation and progression of cancers of different tissues. It seems, therefore, that targeting inhibition of Wnt/β-catenin signaling could be a rational approach for the therapy of cancers of various origins [128]. To this extent, targeting cadherin-17 gene (CDH17) through RNA interference–mediated knockdown inhibited proliferation of both primary and highly metastatic HCC cell lines in vitro and in vivo [129]. Moreover, it was demonstrated that the N-cadherin knockdown (CDH2) led to the inhibition of invasion of human melanoma cells [130]. In addition, the downregulation of N-cadherin lowers the invasiveness of esophageal squamous cell carcinomas in vitro [131].

## 7. Conclusions

Cadherins play an important role in tissue homeostasis and dysfunction. Destabilization of cadherin-catenin complex may result in tumor progression. Cadherins show varied biological functions. The loss of E-cadherin expression during EMT in humans is associated with tumor development and worse prognosis. For these reasons, many studies have shown that E-cadherin can act as tumor suppressor; however, more recently, a role in the cancer progression has been also described.

Many oncogenic signaling pathways modify cadherin cell-cell adhesion. The most commonly described are cyclin kinase inhibitor p27-mediated signaling, MAPK, Ras, Rac1 signaling, PI3K/AKT signaling, as well as Hippo signaling. Abnormal cadherins expression is connected to metastasis, angiogenesis, adhesion and invasion.

Higher expression of N-cadherins is related to tumor aggressiveness, cancer metastasis, apoptosis and angiogenesis in many human and animal cancers.

Overexpression of P-cadherin is also usually a bad prognostic factor, as it is related to tumor progression and invasion and shorter overall survival.

Overall, these adhesion molecules are novel promising targets in cancer treatment, but they may also be useful in predicting a patient’s prognosis in human and veterinary medicine.

## Figures and Tables

**Figure 1 ijms-21-07624-f001:**
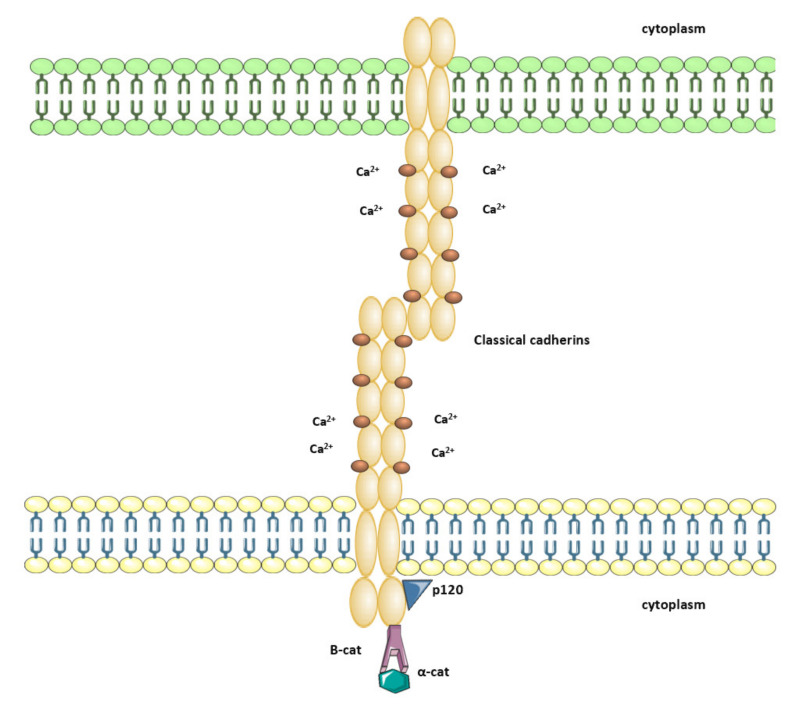
Scheme of the cadherin-catenin complex based on Gama et al. 2012 [29]. Cadherins have extracellular domain, transmembrane domain and a cytoplasmic tail. These transmembrane domains are composed of five cadherin repeats and are responsible for calcium dependent cell-cell adhesion. The cytoplasmic tail is composed of a juxtamembrane domain (proximal to the plasma membrane), which binds to p120-catenin, and a catenin-binding domain, which binds to β-catenin. β-catenin binds to α-catenin, which links cadherins to the actin cytoskeleton.

**Figure 2 ijms-21-07624-f002:**
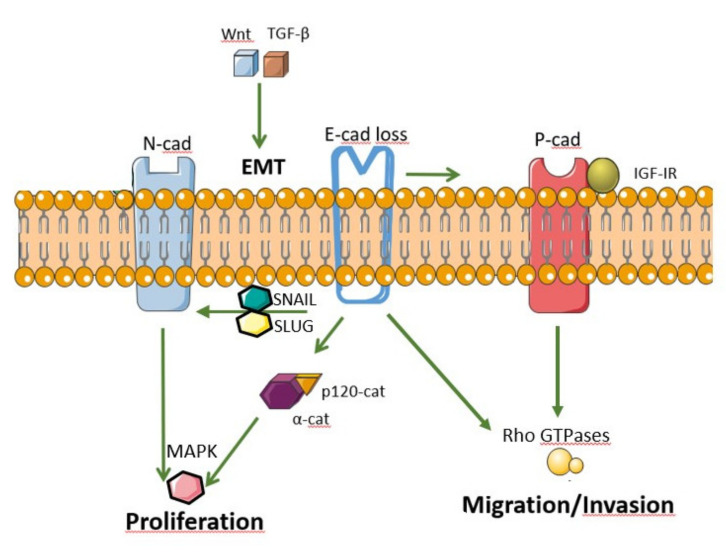
Common signaling pathways of epithelial-to-mesenchymal transition based on Albrecht et al. [80]. Snail and Slug regulate “cadherin switch” by downregulating E-cadherin (E-cad) and inducing the expression of mesenchymal neuronal-cadherin (N-cad). Signaling pathways such as Wnt and TGF-β activate SNAIL and SLUG. N-cad stimulates cell proliferation through mitogen activated protein kinase (MAPK) pathways. The loss of E-cad can also result in the mislocalization of α-catenin and p120 catenin, which leads to the activation of MAPK pathways. Somatic mutations of E-cad leading to its downregulation disrupt normal signaling to Rho GTPases (Rac1 and RhoA), which leads to tumor cell migration and invasion. Upregulated placental-cadherin (P-cad) induces the insulin-like growth factor 1 receptor (IGF-1R) signaling pathway, which also leads to Rho GTPase signaling that promotes migration and invasion.

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
