# Peer review of "Role of Cadherins in Cancer—A Review"

_ijms, 2020, doi:10.3390/ijms21207624_

Round 1

Reviewer 1 Report

The review describes the role of cadherins in cancer.

The authors review a large number of publications. A significant number of the reviewed publications are the literature reviews published earlier. I suggest that the authors include more information using original research papers rather than provide the readers with the review of reviews.

The introduction is too long and unfocused. It contains information about the level of expression of E-cadherin in cancer (lines 70-87). This paragraph will be more relevant for section 5 “Cadherin in human and animal cancer as a prognostic marker”.

Apart from section 5, the information about different cadherins is very rudimentary. Sections 1-4 are dedicated mostly to E-cadherins but not other cadherins. I recommend the authors to include information about other cadherins or revise the title of the review and its sections accordingly.

The text of the review is difficult to read. The current version of the review contains only one figure. The figure is unclear and to understand it the reader should read a long figure legend. I strongly recommend the authors to revise it and make it easier to understand. The review would be improved if authors include more figures. The table summarizing the information provided in section 5 would be useful to include in the review also.

English needs to be checked and slang should be excluded from the publication. For example, the sentence in lines 38-39, the last sentence of the first paragraph: “Recently, it was underscored the relevance of additional 38 cadherins in cancer progression and metastasis, including cadherin 5, 6 and 17”.  First point, there is no information about cadherin 5, 7 and 17 in the review at all.  Why to mention them here? Second point, this sentence is not using English words correctly. 

The usage of terminology should also be revised throughout the text. For example: “calcium-dependent extracellular domains” is incorrect – cadherins meditate Ca-dependent adhesion via their extracellular domains.

Reviewer 2 Report

This review article by Kaszak and coworkers focuses on the role of cadherins in cancer and conveys its message in a brief and straightforward manner. However, the text needs to be greatly improved both in its English form (mistakes, unclear statements and redundant repetitions negatively affect the readability of the paper) and in the scientific discussion of the various issues. Some examples of statements and passages that need revisions are indicated below, but a general and thorough editing of the text is required.

  1. In the Abstract, the statements between line 24 and 27 need revision, it is not clear the connection between the mentioned pathways (“Receptor tyrosine kinase (RTK), Rho GTPases, phosphoinositide 3-kinase (PI3K) and Hippo”) and cadherins.
  2. In the Introduction, the statement at lines 45-46 needs revision: the Authors say here that beta-catenin is the cytoplasmic binding partner as if it is the only one, but as they state in other passages also p120 binds to cadherins.
  3. In the Introduction, lines 78-87, although interesting for their focus on veterinary tumors, are quite confusing, need English revision and a better explanation of the different studies with their sometimes conflicting results.
  4. Introduction line 88 “pacients” is a typo...
  5. In paragraph 2 there are several grammatical errors, such as inappropriate verbal forms after “thus” at lines 100 and 113.
  6. In paragraph 2, lines 100-105 are conflicting with the Introduction (see point 2). Moreover, in the figure legend the Authors say that p120 catenin binds the juxtamembrane domain of E-cadherin, while in others parts of the text they say that p120 catenin binds the distal region and beta-catenin the juxtamembrane domain.
  7. In Paragraph 3, lines 135-137 are confusing and it is not clear which phosphorylations the Authors are describing. Moreover, is beta-catenin released in the cytoplasm also as a consequence of E-cadherin loss?
  8. The signaling pathways mentioned at lines 159-174 are indeed presented as a list and are not clearly explained, therefore the role of cadherins or catenins remains obscure. Please revise this part. Moreover, for the readability of the paragraph it would be better to put together the parts regarding beta-catenin and Wnt signaling.
  9. Lines 223-225: repetition of TGF-beta in both sentences.
  10. Lines 291-293: the abnormal expression of N-cadherin in breast cancer has been already mentioned in the previous sentences of this paragraph.
  11. Another example of a repetitive concept is at line 312-313: indeed, it is redundant to say that E-cadherin is involved in epithelial cellular adhesion, since this has been clearly explained in previous paragraphs.
  12. In Paragraph 5, lines 334-339 are unclear: what do the Authors mean with “cytoplasmic E-cadherin”?
  13. In Paragraph 6, the suggestion made by the Authors at line 368 about MMP inhibitors as activators of E-cadherin is actually intriguing, but the mechanism by which MMPs regulate E-cadherin function should be better explained in the text.
  14. In Paragraph 7, the meaning of the sentence at lines 404-405 is unclear.
  15. Some references are cited in the text with a wrong number: for example, at line 393 the right ref is number 121 and not 122, at line 398 the right ref is number 123 and not 124. The Authors should carefully check the accuracy of references in the text.

Round 2

Reviewer 2 Report

The Authors have successfully improved their review in accordance to Reviewers' suggestions.

A couple of minor points remains to be addressed:

- there are still some errors in some sentences, such as: line 91: "binding" and not "bounding" line 273 and 379: "triggered by" and not "trigger by"  line 886: "of" and not "if"  

- In Paragraph 2, it is still not clear which catenin binds the juxtamembrane domain of cadherins and which the distal part, the following sentences are contradictory and should be changed.

Line 80-81: "The intracellular juxtamembrane region of cadherins binds to β-catenins, while the distal region binds to p120ctns"

Line 86-88: "Specifically, the cytoplasmic tail of cadherin is comprised of juxtamembrane domain, which binds to a family of proteins, including p120catenin (p120; CTNND1)"

Line 91-93: "As mentioned before, cadherins also often bind to β-catenin (CTNNB1) through the terminal catenin-binding domain"

Author Response

This manuscript is a resubmission of an earlier submission. The following is a list of the peer review reports and author responses from that submission.

Round 1

Reviewer 1 Report

I am happy with the Authors's detailed responses and extensive edits to the Review Article. I think it can now move forward for publication.

Reviewer 2 Report

The Authors have addressed many of my concerns, but unfortunately the revised version of the manuscript still maintains some of the defects initially reported.

1) I still find some of the paragraphs quite difficult to read. Some concepts are repeated many times across the manuscript, while others are just swiftly mentioned and not clearly explained (see in particular the final part of Paragraph 4, modified following Reviewer#2 suggestions).

2) The revised part at lines 151-162 is still confusing. In particular, it should be clarified the regulatory role of phosphorylations. In fact, Tyrosine phosphorylation (e.g. mediated by Src) could regulate E-cadherin and its association with beta-catenin; whereas, after the latter is released in the cytoplasm, it becomes phosphorylated in Serine/Threonine by GSK3beta (in absence of Wnt signaling), which triggers its subsequent proteasomal degradation. So, while the first mechanism leads to beta-catenin signaling activation (and transcriptional impact), the second is an inactivating mechanism.

3) The conclusion of  chapter 5 (lines 411-417) is rather unsatisfactory,  being generic and poorly informative. I would recommend its improvement, by highlighting main emerging concepts and open questions.

4) The text at points should be rephrased. As an example, see a new statement added in the revised version at line 42: “Recently, the relevance of other group of cadherins, including cadherin 5, 6 and 17 in cancer progression and metastasis has also discussed [4]”. Maybe change to: “Recently, it was underscored the relevance of additional cadherins in cancer progression and metastasis, including cadherin 5, 6 and 17 [4]”.

Also, at line 51: “The cytoplasmic binding partner is beta-catenin but depending on the subtype of cadherin they may bind to plakoglobin as well”. Maybe change to “… but certain cadherins may also bind plakoglobin”.

Moreover, there are still a number of typos, such as:

Line 27: RKT=RTK

Line 59: relay=rely

Line 114, “further” should actually be “farther”

Line 205: RhoGPTase=RhoGTPase

Line 274: can associated=can be associated

Round 2

Reviewer 2 Report

The Authors have addressed most of my concerns. Some of the new edited text however is in need of English revision. In one case, they should improve the description integrating my comments, not simply appending them to the old text. I recommend that whatever final editing of the manuscipt is then revised by an expert English language revisor.

Specific points:

Lines 109-111: the edited statement is confusing; I suggest “decreased expression of E-cadherin is associated with tumor malignancy and metastatic progression, which leads to shorter patient survival”.

Lines 183-205: the authors cannot improve this part simply by appending my comments; the entire paragraph should be rephrased for clarity, integrating all information as appropriate. Moreover the new edited statement in lines 183-185 is confusing.

Line 280: the edited text requires language revision; I suggest “The main transcriptional activators of EMT are Wnt/β-catenin, TGF-β, and Hedgehog signaling pathways [59,60,63,64]. In addition, signaling molecules such as ILK, FAK and SRC, TGF-β have an impact on the activation of EMT[65].

Lines 297-301: it was better the previous text (which has been deleted and rephrased); I recommend rejecting changes and reverting to previous version.

Line 408: it was better the previous text (which has been deleted and rephrased); I recommend rejecting changes and reverting to previous version.
